# A Small Step, a Giant Leap: Somatic Hypermutation of a Single Amino Acid Leads to Anti-La Autoreactivity

**DOI:** 10.3390/ijms222112046

**Published:** 2021-11-07

**Authors:** Tabea Bartsch, Claudia Arndt, Liliana R. Loureiro, Alexandra Kegler, Edinson Puentes-Cala, Javier Andrés Soto, Biji T. Kurien, Anja Feldmann, Nicole Berndt, Michael P. Bachmann

**Affiliations:** 1Department of Radioimmunology, Institute of Radiopharmaceutical Cancer Research, Helmholtz-Zentrum Dresden-Rossendorf (HZDR), 01328 Dresden, Germany; t.bartsch@hzdr.de (T.B.); c.arndt@hzdr.de (C.A.); l.loureiro@hzdr.de (L.R.L.); a.kegler@hzdr.de (A.K.); epuentes@corrosion.uis.edu.co (E.P.-C.); jav.soto@mail.udes.edu.co (J.A.S.); a.feldmann@hzdr.de (A.F.); n.berndt@hzdr.de (N.B.); 2Corporación para la Investigación de la Corrosión (CIC), Piedecuesta 681011, Colombia; 3BIOGEN Research Group, University of Santander, Faculty of Health Sciences, Cúcuta 540001, Colombia; 4The Arthritis and Clinical Immunology Program, Oklahoma Medical Research Foundation, University of Oklahoma Health Sciences Center, Oklahoma City, OK 73104, USA; Biji-Kurien@omrf.org; 5Tumor Immunology, University Cancer Center (UCC), University Hospital Carl Gustav Carus Dresden, TU Dresden, 01307 Dresden, Germany; 6National Center for Tumor Diseases (NCT), 03128 Dresden, Germany

**Keywords:** anti-La/SS-B antibodies, autoimmunity, La/SS-B autoantigen, systemic lupus erythematosus, primary Sjögren’s syndrome

## Abstract

The anti-La mab 312B, which was established by hybridoma technology from human-La transgenic mice after adoptive transfer of anti-human La T cells, immunoprecipitates both native eukaryotic human and murine La protein. Therefore, it represents a true anti-La autoantibody. During maturation, the anti-La mab 312B acquired somatic hypermutations (SHMs) which resulted in the replacement of four aa in the complementarity determining regions (CDR) and seven aa in the framework regions. The recombinant derivative of the anti-La mab 312B in which all the SHMs were corrected to the germline sequence failed to recognize the La antigen. We therefore wanted to learn which SHM(s) is (are) responsible for anti-La autoreactivity. Humanization of the 312B ab by grafting its CDR regions to a human Ig backbone confirms that the CDR sequences are mainly responsible for anti-La autoreactivity. Finally, we identified that a single amino acid replacement (D > Y) in the germline sequence of the CDR3 region of the heavy chain of the anti-La mab 312B is sufficient for anti-La autoreactivity.

## 1. Introduction

The nuclear autoantigen La, also known as Sjögrens’ syndrome associated antigen B (SS-B), is highly conserved during evolution, including between humans and rodents (for a sequence comparison, see Appendix A). Autoantibodies to the nuclear autoantigen La/SS-B are frequently found in sera of patients with systemic lupus erythematosus (SLE) and primary Sjögrens’ syndrome (pSS) [1]. In longitudinal studies, anti-La antibodies (abs) have been described as the first detectable autoantibodies [2]. So far, different mechanisms have been put forward for the development of autoimmunity in SLE and pSS patients, including for example, epitope spreading and molecular mimicry [3,4,5,6,7,8,9]. Furthermore, an impaired clearance of apoptotic cell material or an immune response to hidden or post-translationally modified epitopes was postulated for the generation of autoimmunity [10,11,12]. In autoimmune-prone mice, Guo et al. observed that abs to nuclear antigens (ANAs) can occur as the result of somatic hypermutation (SHM) of originally non-autoreactive B cells [13]. In this mouse model, the SHMs were shown to occur in germinal centers. However, it remained unclear whether or not the SHMs were dependent on T cell help, and it was not determined which amino acid (aa) replacements finally led to autoreactivity.

Until now, many groups have tried to elicit monoclonal antibodies (mabs) to nuclear antigens, including to the La/SS-B antigen [14,15,16,17,18,19,20,21,22,23,24,25]. With few exceptions, detailed epitope mapping data are missing for most of the anti-La mabs. The most frequently used anti-La mab, SW5, is directed to a discontinuous epitope (aa 112–138 and aa 171–183) of human La protein [19,20,21]. It immunoprecipitates human La protein from total extracts of human cells but not mouse La protein from total extracts of murine cells. After SDS-PAGE and immunoblotting under denaturing conditions, it still recognizes human La protein and fails to react with mouse La protein. In contrast to the eukaryotically expressed mouse La protein, it recognizes mouse La protein if recombinantly expressed in *E. coli*. Therefore, the epitope region recognized by the anti-La mab SW5 should be post-translationally modified in native eukaryotic mouse La protein [26]. In contrast to the anti-La mab SW5, the anti-La mabs 5B9 and 7B6 react with short continuous aa sequences. Neither the anti-La mab 5B9 nor the anti-La mab 7B6 immunoprecipitates La protein from total extracts of human or murine cells. Therefore, both peptide epitopes are cryptic and not accessible in native eukaryotic human La protein [26]. The anti-La mab 5B9 recognizes the aa sequence KPLPEVTDEY (aa 95–104 of human La protein), which is a part of the random coiled “wing like” region in the N-terminal domain of the La protein connecting the La motif with the RNA recognition motif (RRM) 1 [26]. The anti-La mab 7B6 recognizes the aa sequence EKEALKKIIEDQQESLNK (aa 311–328 of human La protein). This aa sequence is part of the α3 helix in the RRM2 of La protein [24,26]. The epitope is thereby part of a previously described nuclear retention element and potential dimerization domain. According to recent data, La protein is a redox sensor that undergoes conformational changes in a redox-dependent manner whereby the human epitope region recognized by the anti-La mab 7B6 becomes accessible [26,27]. The murine counter part of the 7B6 epitope is post-translationally modified and is not recognized by the anti-La mab 7B6. In a series of studies (e.g., [27,28,29,30,31,32,33,34,35,36,37,38,39,40,41,42,43]), both the 5B9 and the 7B6 La epitopes have been used as peptide tags and thereby verified.

At least to our knowledge, besides these well characterized anti-La mabs, none of the other previously described anti-La mabs cross-reacts with native eukaryotic mouse La protein. Consequently, none of the known anti-La mabs is a bona fide anti-La autoantibody. Most recently, however, we described novel anti-La mabs. Among them is the anti-La mab 312B, which co-precipitates native eukaryotic human and murine La protein [26] (see also below). The ab recognizes a discontinuous epitope present in the La motif (for a schematic view, see also Appendix A) which is sensitive to oxidoreduction [26,27]. 

As deduced from the primary aa sequence, the anti-La mab 312B underwent SHM (see also below) which resulted in 11 aa replacements. When we reverted all the SHMs in the primary aa sequence of the autoreactive anti-La mab 312B back to its germline sequence, the resulting ab no longer reacted with La protein [27] (see also Figure 1). Bearing in mind that the germline 312B B cell was allowed to leave the bone marrow after successful recombination, we expect that the ab was most likely not autoreactive from the beginning. Consequently, these data led us to the conclusion that autoreactivity can be acquired during B cell maturation due to T cell-dependent SHMs, including in tolerant mice. In order to confirm this idea, we had to answer the obvious remaining question: which of the SHM(s) finally converted the non-autoreactive B cell to the B cell secreting the autoreactive anti-La ab? Here, we show that the replacement of just one aa—an aspartate residue in the complementarity determining region (CDR) 3 of the variable heavy chain domain (V_H_) with a tyrosine residue—restores the anti-La autoreactivity of the non-anti-La reactive germline 312B derivative. 

## 2. Results and Discussion

### 2.1. The CDRs of the Anti-La Mab 312B Are Relevant for Anti-La Reactivity

In a recent study, we were able to show that the replacement of all SHMs acquired in the primary aa sequence of the anti-La mab 312B back to its predicted germline sequence results in a loss of anti-La reactivity [26]. In order to facilitate this analysis, we cloned and expressed two single-chain fragment variables (scFvs) of the respective mature and germline variable domains of the 312B heavy (V_H_) and light chain (V_L_) sequences, which were then fused to the Fc domain of human IgG4 molecules [26]. Figure 1A shows a schematic view of this ab derivative. Such recombinant ab derivatives keep their specificity and most of their binding affinity compared to the original mab (see also below). Both constructs were transduced into the murine 3T3 cell line for permanent ab production. Both recombinant abs were purified from cell culture supernatants using protein A affinity chromatography. As shown in Figure 1B, both constructs encoding either the mature 312B ab binding domains (Figure 1B, 312B) or the predicted 312B germline ab binding domains (Figure 1B, germline 312B) are well expressed. 

It is commonly accepted that the CDRs of an ab form the paratope, which binds to the epitope of the antigen. In order to confirm that the CDRs of the anti-La mab 312B are indeed responsible for its anti-La reactivity we grafted them to human framework regions. For this purpose, the murine CDRs of the anti-La mab 312B were in silico fused to the best fitting human framework, resulting in the humanized 312B V_H_ and V_L_ sequences. Based on these domains, an additional scFv was constructed and fused to the human IgG4-Fc domain (Figure 1A). The obtained humanized 312B ab construct was permanently transduced into murine 3T3 cells. After purification from cell culture supernatant via protein A affinity chromatography, humanized 312B ab was analyzed by SDS-PAGE and subsequent Coomassie-Brilliant Blue staining (Figure 1B, hu312B). 

Next we compared the anti-La reactivity of the recombinant murine 312B ab (Figure 1C, 312B), the germline 312B ab (Figure 1C, gl312B), and the humanized 312B ab constructs (Figure 1C, hu312B) by SDS-PAGE/immunoblotting. Besides their reactivity to recombinant human La (Figure 1C, rh-La), we tested their reactivity against a mutant La protein version in which the three cysteine residues present in the primary aa sequence were mutated to alanine. This triple cysteine mutant was termed TCM-La (Figure 1C, TCM-La) [27]. From previous studies, we know that the replacement of the three cysteine residues makes La protein insensitive to oxidation [27]. The major reason to test both La protein versions was as follows. After description of the anti-La mab 312B, we learned that La protein is sensitive to oxidation [27]. Oxidation leads to a conformational change of the La motif, which also effects the reactivity of the anti-La mab 312B, showing a higher binding affinity towards the permanently reduced TCM-La. Therefore, we wanted to rule out that the lack of anti-La reactivity of the germline 312B ab is somehow related to such a redox-dependent conformational effect and decided to compare the reactivity of all the 312B ab derivatives against recombinant human wildtype La protein (Figure 1C, rh-La) and the triple cysteine La mutant (Figure 1C, TCM-La). As shown in Figure 1C, the murine 312B ab (Figure 1C, 312B, rh-La, TCM-La) and the humanized 312B ab (Figure 1C, hu312B, rh-La, TCM-La) react with both La proteins, while the germline 312B construct does not recognize either wildtype La protein (Figure 1C, gl312B, rh-La) or the mutant La protein (Figure 1C, gl312B, TCM-La). Consequently, the lack of anti-La reactivity of the germline 312B ab is not related to oxidoreduction. Most importantly, these data show that the CDRs of the anti-La mab 312B can transfer anti-La reactivity; thus, we can confirm that the CDRs of the anti-La mab 312B are responsible and sufficient for its anti-La reactivity while the aa replacements in the framework regions may be less important for anti-La reactivity. 

In order to support the immunoblotting data, we determined the apparent K_D_ values of the recombinant abs by ELISA (for a comparison of the binding curves, see also below). Bearing in mind that wildtype La protein always represents an undefined, varying mixture of fully and partially oxidized as well as reduced La protein, the K_D_ values were estimated using the TCM-La protein, which is no more sensitive to oxidation. We estimated a K_D_ value of 0.5 (±0.4) nmolar for the original anti-La mab 312B. For the murine scFv-based IgG4 derivative (312B), which was constructed as schematically summarized in Figure 1A, we measured a K_D_ value of 0.3 (±0.2) nmolar. Consequently, the recombinant derivative shows no loss of affinity compared to the original hybridoma-derived mab, but even a slightly improved binding. For the humanized version of the 312B construct, we determined a K_D_ value of 1.5 (±1) nmolar. Thus, there is a minor loss of affinity during humanization, which is, however, in the same range as observed in previous humanization approaches. The remaining affinity is still in the range of a high affinity ab. For the germline 312B construct, we could not estimate any K_D_ value at all. In summary, these data confirm the immunoblotting data and show that the recombinant scFv-based IgG4 construct has comparable binding capability to the original hybridoma ab.

### 2.2. A Closer Look at the Primary Sequence of the Anti-La Mab 312B

As indicated by the somatic hypermutations (SHMs) present in both the V_H_ and V_L_ of the anti-La mab 312B (Figure 2: germline sequence, red; mature murine 312B sequence, blue), the adoptively transferred anti-human La T cells somehow provided help to the germline 312B B cell. This is in line with the failure to obtain anti-La hybridomas from human La transgenic mice after adoptive transfer of T cells from non-immunized animals [26]. Besides SHM, there is also evidence for the editing of the V_H_ sequence as indicated by the presence of a series of N-nucleotides between the V and the D gene elements that were used during recombination of the 312B V_H_ gene. Due to the random nature of the N-nucleotide insertions and the not exactly predefined 5’-start and 3’-end of the D-element used during the VDJ recombination step, there will always remain some uncertainty about the exact origin of the respective VDJ gene sequence. Based on the nucleotide sequence of the 312B V_H_ domain, we currently favor the following course of events that happened during recombination of the 312B V_H_ gene elements. During the first DJ recombination step of the used D gene element (IGHD2-3*01 or DSP2.9) with the used J element (IGHJ2*01), the D element was truncated at the 3’-end (**TCTATGATGGT**TACTAC) whereby the two triplets (TACTAC) were deleted. Fusion of the resulting DJ-element to the V-element (IGHV1-9 according to NCBI and IMGT/DomainGapAlign) resulted in an unproductive out-of-frame V_H_ domain. The reading frame was rescued by a relatively long insertion of 10 N-nucleotides (TCAAGGTCTA) encoding the aa sequence SRS (Figure 2: heavy chain, dashed line) between the V and the DJ elements. Of course, the insertion of the N-nucleotides also determined the reading frame of the DJ element and also the isoleucine following the aa sequence SRS. It is noteworthy that we kept these N-nucleotides in our germline construct because otherwise, the V_H_ sequence could not be translated into a functional heavy chain. For this reason, the N-nucleotides are present in the non-anti-La reactive 312B germline B cell sequence. As this germline encoded ab fails to react with La protein, the N-nucleotides are less important for anti-La reactivity. Of course, we have to keep in mind that these N-nucleotides determined the used reading frame of the DJ element and thereby the aa sequence of the CDR3 region. After the successful recombination of the 312B V_H_ gene, the 312B B cell successfully recombined the light chain. The resulting germline 312B B cell was able to produce an ab. However, this germline ab did not react with La protein as shown recently [26] (see also Figure 1C, gl312B and below). Most likely, it was also not autoreactive as it was not eliminated but able to leave the bone marrow. 

Following this view, the non-anti-La reactive germline ab-expressing 312B B cell was attracted by the adoptively transferred activated T cells to the lymph node (spleen). The presence of SHMs in the mature 312B B cell suggests that the adoptively transferred T cells provided help. Therefore, we believe that a (first) mutation occurred which accidentally converted the reactivity of the germline B cell to an anti-La (auto)reactive B cell. Thanks to the presence of the adoptively transferred anti-La helper T cells, the immature B cell received further help which supported additional SHM(s) in the V_H_ domain, finally leading to the anti-La autoreactive 312B B cell. To confirm this hypothesis, we had to unravel the SHM(s) that caused this anti-La autoreactivity.

Overall, the primary sequence of the V_H_ domain of the anti-La mab 312B contains 11 aa replacements compared to the predicted germline sequence. As shown in Figure 2, seven aa replacements occurred in the framework regions, with only four aa replacements in the CDR regions. According to the CDR grafting experiment, one of the four aa replacements in the CDR regions should be key for anti-La autoreactivity. Looking at these four mutations in the CDR regions, the replacement of an aspartate residue to a tyrosine residue (Figure 2, red box) came into our focus, which happened in the CDR3 region of the V_H_ domain. 

It is generally accepted that the CDR3 region of the V_H_ domain has the highest contribution to the variability and therefore specificity of an ab. The additional tyrosine residue further increases the already high content of tyrosine residues in the primary aa sequence of the anti-La mab 312B, including in its CDR3 region. It is noteworthy that the high content of tyrosine residues is not unique for the anti-La mab 312B. From our previous studies, the primary aa sequences of 11 anti-La mabs are available, including those of the anti-La mabs SW5, 5B9, and 7B6, which were obtained by three independent hyperimmunization experiments [26]. In addition, we determined the primary aa sequences of the recently described anti-La mabs 312B, 2F9, 32A, 27E, 24BG7, 22A, 16C, and 13C5B, which were obtained after adoptive transfer of anti-human La T cells to a human La transgenic mouse [26]. When looking at these primary aa sequences, the relatively high number of tyrosine residues (including in the CDRs) of all these anti-La abs is striking. It is well known that in the tertiary structure of an ab, all CDRs are brought into close vicinity to form the paratope region. Thus, all the tyrosine residues in the anti-La abs (at least the ones in their CDR regions) are brought into close vicinity during the three dimensional folding of the primary heavy- and light-chain protein sequences. A closer look at the tyrosine residue content shows that the overall amount of tyrosine residues in these anti-La abs ranges from 10 to 21 tyrosine residues. Both the anti-La mabs SW5 and 7B6, which are specific for human La but not for mouse La protein, contain 12 tyrosine residues. Five of these 12 tyrosine residues are part of the CDR regions, and seven of them are part of the framework region. It is noteworthy that of the seven tyrosine residues present in the framework regions, six of them are adjacent to CDRs, meaning that actually 11 of the 12 tyrosine residues are either directly in or close to the CDR regions. With 15 tyrosine residues, the anti-La mab 5B9, which recognizes a continuous but cryptic epitope on both human and mouse La protein, even contains three more tyrosine residues. As described recently, all the IgG type anti-La mabs (312B, 2F9, 32A, 27E, 24BG7, and 22A) which were obtained after the adoptive transfer of anti-human La T cells recognize discontinuous epitopes. With two exceptions, all these anti-La mabs contain an even higher amount of tyrosine residues: they contain between 17 and 19 tyrosine residues, of which 9–10 tyrosine residues are present in the CDRs. If we include in this calculation those tyrosine residues adjacent to the CDR regions, then even 14–17 of the tyrosine residues are either in or close to the CDR regions. In the case of the anti-La mab 312B, 9 of its 18 tyrosine residues (or even 17 if the adjacent tyrosine residues are included) are in (or close to) the CDRs. It is noteworthy that all tyrosine residues except for the additional one in the CDR3 region (Figure 2, red box) already exist in the germline sequence (Figure 2, tyrosine residues highlighted in yellow). We therefore expected that these tyrosine residues might somehow play a role in the anti-La reactivity, and therefore, the additional tyrosine residue caused by SHM of the aspartate could play a key role in anti-La reactivity.

At a first glance, the extremely high content of tyrosine residues in the IgM type anti-La ab 13C5B seems to argue against this idea. With 21 tyrosine residues, of which 10 tyrosine residues are part of the framework regions and 11 tyrosine residues are part of the CDRs, this polyreactive IgM type anti-La mab 13C5B contains the highest number of tyrosine residues of all anti-La mabs known so far. However, according to our recent sequence analyses, this polyreactive IgM type anti-La mab should not represent a precursor of the IgG type anti-La mabs. Therefore, we speculate that the anti-La reactivity of the 13C5B mab is part of its polyreactivity: the presence of the high number of tyrosine residues may simply increase the chance that those tyrosine residues, which are required for anti-La reactivity, are also present in the primary sequence.

### 2.3. The Lucky Punch

Bearing in mind the high number of tyrosine residues in anti-La mabs on the one hand, and the replacement of the aspartate residue in the CDR3 region of the V_H_ domain to a tyrosine residue on the other hand, we therefore decided to start our mutational analysis by constructing a germline 312B derivative in which the aspartate residue is replaced by tyrosine. This mutant 312B ab was termed gl312B-D > Y (germline mutant in which the aspartate residue in the CDR3 region of the V_H_ domain is replaced by a tyrosine residue). To facilitate its cloning and expression, we followed again the above described strategy: We constructed a scFv consisting of the mutated germline 312B V_H_ and V_L_ sequences, which was then fused to the human IgG4-Fc domain. In this mutated germline variant, the V_H_ domain contained the D to Y mutation in the CDR3 region. The resulting construct was again permanently transduced into eukaryotic cells. Secreted recombinant abs were purified via protein A affinity chromatography from cell culture supernatants. The isolated abs were analyzed by SDS-PAGE and stained with Coomassie-Brilliant Blue (Figure 3A). The gl312B-D > Y mutant ab was tested for anti-La reactivity using SDS-PAGE and immunoblotting (Figure 3B). Luckily, the single aa replacement already restored the anti-La reactivity to both wildtype human La protein (Figure 3B, gl312B-D > Y, rh-La) and the TCM-La mutant protein (Figure 3B, gl312B-D > Y, TCM-La). 

The SDS-PAGE/immunoblotting experiment shown in Figure 3 was performed under denaturing conditions, meaning that the analyzed La proteins were heat denatured prior to SDS-PAGE. Thus, it was not determined whether the gl312B-D > Y mutant ab also recognizes native La protein such as the mature murine 312B ab. To answer this question, we performed a co-immunoprecipitation experiment using a total extract of the human HeLa cell line. As shown in Figure 4, the gl312B-D > Y mutant (Figure 4, gl312B-D > Y) precipitates native La protein such as the murine (Figure 4, 312B) and the humanized 312B ab (Figure 4, hu312B) while, as expected, the germline 312B fails to precipitate native La protein present in the total extract (Figure 4, germline 312B). Consequently, the single D > Y aa replacement in the CDR3 region of the VH domain is sufficient to directly confer anti-La autoreactivity to the non-anti-La reactive germline 312B ab in just one step.

In order to get initial information about the contribution of the aspartate to tyrosine mutation on the overall affinity of the 312B ab, we estimated the apparent K_D_ value of the gl312B-D > Y ab derivative (Figure 5). As mentioned above, to avoid artefacts by oxidation, the K_D_ value was determined using the TCM-La mutant protein. We thereby measured a K_D_ value of 2.0 (±1.6) nmolar, which is in the same range as the affinity of the humanized ab hu312B. Thus, it is slightly reduced compared to the mature, murine 312B ab but already in the range of a high affinity ab. As one can expect, the additional SHMs in the primary sequence of the mature 312B ab have further improved its binding capability, but the aspartate to tyrosine mutation was already sufficient for high-affinity anti-La reactivity. 

This interpretation is also supported by comparison of the binding curves obtained for the original hybridoma anti-La mab 312B (Figure 5, anti-La 312B mab); the murine 312B IgG4 construct (Figure 5, 312B); the germline IgG4 construct (Figure 5, gl312B); the humanized 312B IgG4 construct (Figure 5, hu312B); and the germline mutant IgG4 construct, in which the aspartate residue is mutated to a tyrosine residue (Figure 5, gl312B-D > Y). As shown in Figure 5, the anti-La mab 312B produced from the original hybridoma (Figure 5, anti-La 312B mab) shows similar binding capability to the corresponding recombinant murine 312B IgG4 ab (Figure 5, 312B). Again the 312B derivative in which all the SHMs were mutated back to the germline sequence does not show any binding (Figure 5, gl312B). There is no binding at all, even at high ab concentrations. In contrast, the replacement of the aspartate residue to the tyrosine residue in the CDR3 region of the V_H_ domain restores the binding capability (Figure 5, gl312B-D > Y). Interestingly, the binding curve of the humanized 312B ab (Figure 5*,* hu312B) is almost identical to the germline 312B-D > Y mutant. These data suggest that the D to Y replacement was the key step which converted the non-anti-La reactive germline 312B B cell to the autoreactive anti-La 312B B cell while the additional SHMs only further increased its affinity.

In summary, here, we present for the first time (to our knowledge) experimental evidence that a single aa replacement can convert a non-autoreactive B cell into an autoreactive one. The replacement and further maturation may be triggered by help via activated T cells. Bearing in mind that La protein can form complexes with viral nucleic acids, including for example, small viral RNAs (e.g., EBER or VA RNAs [44,45]), but also with viral mRNAs or genomic RNA (e.g., poliovirus RNA, Hepatitis C; e.g., [46]) such an accident in combination by chance may also occur during an immune response against an infectious agent or nucleic acid La complex: T helper cells may accidentally provide help to non-anti-La B cells, accidentally triggering an SHM which converts the B cell into an autoreactive B cell.

## 3. Materials and Methods

### 3.1. Recombinant Human La Protein Expression and Characterization

Recombinant human La protein (rh-La) and the triple cysteine mutant (TCM-La) were expressed as described in [27]. The proteins were subsequently purified from bacteria lysates via their 6xHis-tag using Ni-NTA affinity chromatography. Purity and concentration of the proteins was analyzed by SDS-PAGE and subsequent Western Blotting [47] or Coomassie-Brilliant Blue G250 staining [48].

### 3.2. Construction, Expression, and Purification of Recombinant 312B Antibody Derivatives

Based on the V_H_ and V_L_ sequences of the anti-La mab 312B [26], four different recombinant IgG4 ab constructs were created: murine 312B-IgG4 (312B); humanized 312B-IgG4 (hu 312B); germline 312B-IgG4 (germline 312B); and the mutated germline 312B-IgG4 (gl312B-D > Y), which carries an aspartate to tyrosine mutation in the CDR3 region of the V_H_ domain. Identification of the murine 312B V_H_ and V_L_ sequences and comparison against germline V_H_ and V_L_ sequences was performed using the NCBI and IMTG data library as previously published [26]. The mutated germline V_H_ domain (D > Y) was generated by replacing the aspartate (D) residue in the CDR3 region in the germline sequence with a tyrosine (Y) residue. For humanization of the V_H_ and V_L_ domains, homologous human V_H_ and V_L_ sequences were identified using the NCBI IgBlast sequence analysis and the IMGT/DomainGapAlign tool [49,50]. Annotations of the FWRs and CDRs were done according to the Kabat [51] and IMTG database. Based on the best fitting human FWRs and the murine CDRs, humanized 312B V_H_ and V_L_ sequences were designed in silico. Cloning of all four 312B-IgG4 constructs was performed according to a previously published strategy [40,41]; construction of the murine 312B-IgG4 (312B) and germline 312B-IgG4 (gl312B) were already detailed in [26]. Briefly, the respective murine, humanized, germline, or mutated germline variable domains were in silico fused in V_H_–V_L_ orientation via flexible linkers consisting of three G_4_S motifs. Corresponding DNA sequences were purchased from Eurofins Genomics (Ebersbach, Germany) and subsequently cloned via *Sfi*I/*Mre*I restriction sites upstream of the CH_2_ and CH_3_ domains of human IgG4 abs in the lentiviral expression vector p6NST50. Plasmids were subsequently used to establish permanent 3T3 production cell lines by lentiviral transduction. All recombinant IgG4 derivatives possess an N-terminal murine Igκ leader sequence promoting secretion of abs. Recombinant IgG4 constructs were purified from cell culture supernatants using the Protein A HP Spin Trap (Sigma-Aldrich Chemie GmbH, Steinheim, Germany), according to the manufacturer’s instructions. The concentration and purity of the abs were determined with SDS-PAGE and Coomassie-Brilliant Blue staining or Western Blotting as described above. Human and murine specific sequences of La protein were described previously [52]. 

### 3.3. Immunoblotting

Two µg of rh-La or TCM-La protein were separated via SDS-PAGE and subsequently transferred onto a nitrocellulose membrane by Western Blotting. After blocking, the membrane was incubated with the respective IgG4 ab constructs (312B/gl312B/hu312B/gl312B-D > Y) using ab dilutions of 1.5 µg/mL in Blocking Solution (DIG wash and block buffer set, Roche Diagnostics GmbH, Mannheim, Germany). Ab binding was detected using anti-human IgG4-horseradish peroxidase (HRP) secondary ab (Southern Biotech, BIOZOL Diagnostica Vertrieb GmbH, Eching, Germany).

### 3.4. Immunoprecipitation

For preparation of total cell extracts (TE), HeLa cells were resuspended in Lysis Buffer (50 nM Tris/HCl, pH, 8.0, 150 mM NaCl, 1% IGEPAL^®^ CA-630 (CAS 9002-93-1, Santa Cruz Biotechnology, Inc., Heidelberg, Germany)), incubated on ice for 10 min, and then centrifuged at 10,000 × *g*. The supernatant (total extract, TE) was subsequently used for the immunoprecipitation and as positive control (POS). Therefore, the TE was mixed with 2 µg of purified ab or PBS (negative control, NEG) and Protein G MicroBeads (Miltenyi Biotec B.V. & Co. KG, Bergisch Gladbach, Germany) and incubated on ice for 30 min. The magnetically labeled beads were then separated from TE using magnetic µColumns (Miltenyi Biotec B.V. & Co. KG, Bergisch Gladbach, Germany) and eluted from the columns with heated Elution Buffer (50 mM Tris/HCl, pH 6.8, 50 mM DTT, 1% SDS, 40 µg/mL bromophenol blue, 10% Glycerin). Afterwards, co-precipitated immune complexes were analyzed by SDS-PAGE and immunoblotting. La protein was detected using anti-5B9 mab and TrueBlot^®^ anti-mouse IgG-HRP ab (Rockland Immunochemicals Inc., Gilbertsville, PA, USA).

### 3.5. ELISA

The BD OptiEIA™ Reagent Set B (BD Biosciences Pharmingen, Heidelberg, Germany, #550534) was used for ELISA. A number of 96-well plates (Sigma-Aldrich Chemie GmbH, Steinheim, Germany, #CLS3590-100EA) were coated with 0.3 µg TCM-La per well at 4 °C overnight. After blocking, increasing concentrations of the respective abs were added and incubated for 1 h at 37 °C. The bound recombinant abs were detected with an anti-human IgG-HRP ab (Sigma-Aldrich Chemie GmbH, Steinheim, Germany). After addition of the substrate solution (NanoQuant infinite M200 Pro, Tecan Group AG), OD was measured at 450 nm. K_D_ values and statistical evaluation were calculated with GraphPad Prism 9 software (GraphPad Prism Inc., La Jolla, CA, USA).

## Figures and Tables

**Figure 1 ijms-22-12046-f001:**
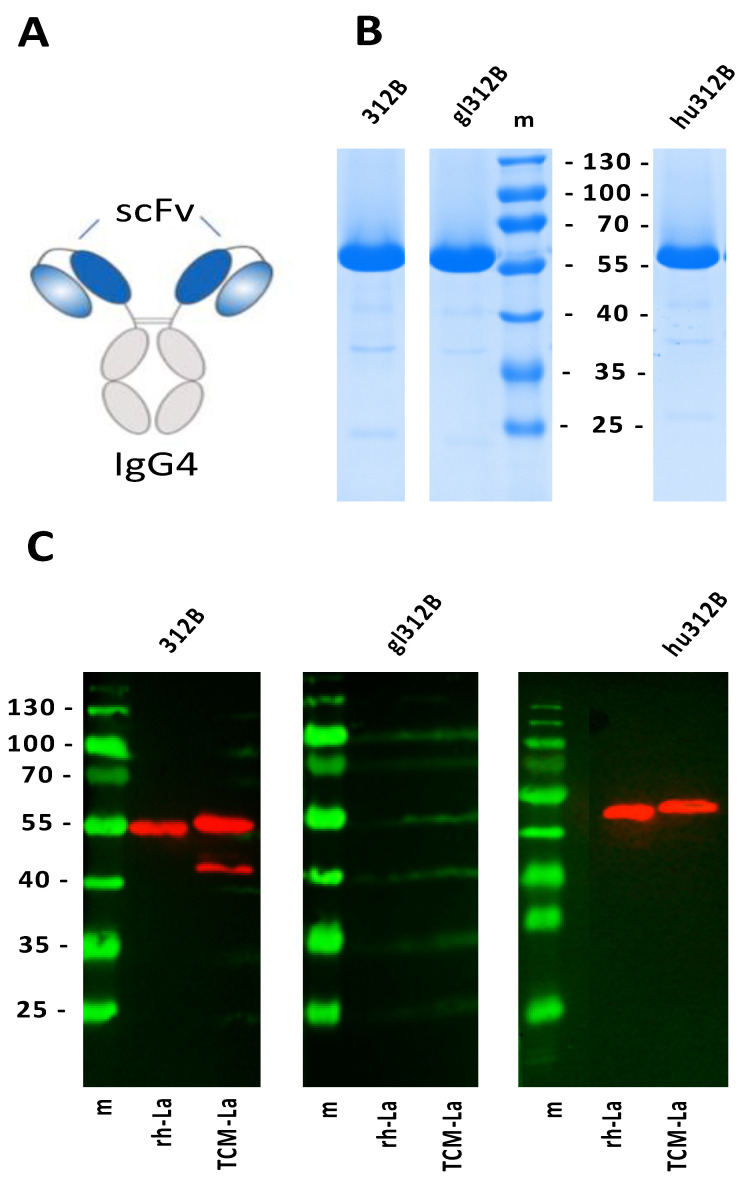
The Complementarity Determining Regions (CDRs) of the anti-La mab 312B confers anti-La reactivity. (**A**) Schematic representation of recombinant 312B ab constructs. Based on the murine (mature), germline, and humanized variable heavy and light chain domains of the anti-La mab 312B, different single-chain fragment variables (scFvs) were constructed and fused to the human IgG4-Fc domain to obtain recombinant murine (mature), germline, or humanized 312B ab constructs. (**B**) Recombinant 312B constructs were isolated by protein A affinity chromatography from cell culture supernatant of permanent 3T3 production cell lines. Purified proteins (312B: mature ab; gl312B: all SHMs were mutated back to the germline sequence; hu312B: CDRs of the murine 312B ab were grafted to the best fitting human framework regions) were separated by SDS-PAGE and subsequently stained with Coomassie-Brilliant Blue. (**C**) The purified 312B ab constructs were tested by SDS-PAGE/immunoblotting against human recombinant La protein (rh-La) and a mutant La version in which the three cysteine residues were mutated to alanine (TCM-La), which makes La protein insensitive to oxidation. m, protein ladder (kDa).

**Figure 2 ijms-22-12046-f002:**
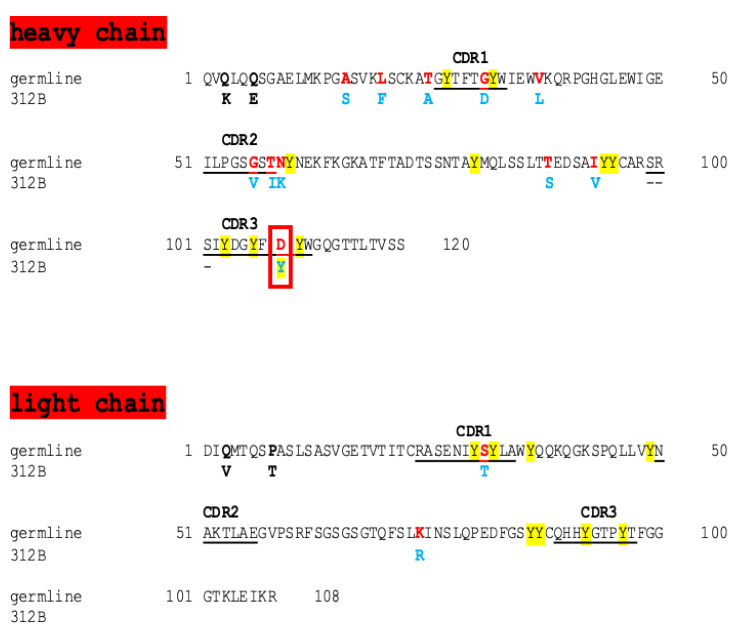
Comparison of the primary V_H_ and V_L_ aa sequence of the anti-La mab 312B with its germline sequence. The aa mutated in the germline sequence by somatic hypermutation (SHM) are labeled in red and those replaced by the aa are labeled in blue for the mature, murine 312B V_H_ and V_L_ aa sequences. The aa given in bold are most likely caused by the degenerated PCR primers used for amplification of the V_H_ and V_L_ domains. The CDRs are underlined. All tyrosine residues are highlighted in yellow. Red box: the only tyrosine residue which is not encoded by the germline sequence is caused by a SHM leading to replacement of the aspartate residue. Dashed line in the CDR3 region: during recombination of the V_H_ gene, N-nucleotides were inserted between the V and the DJ gene elements. The inserted N-nucleotides encode the aa SRS in the CDR3 region of the V_H_ domain.

**Figure 3 ijms-22-12046-f003:**
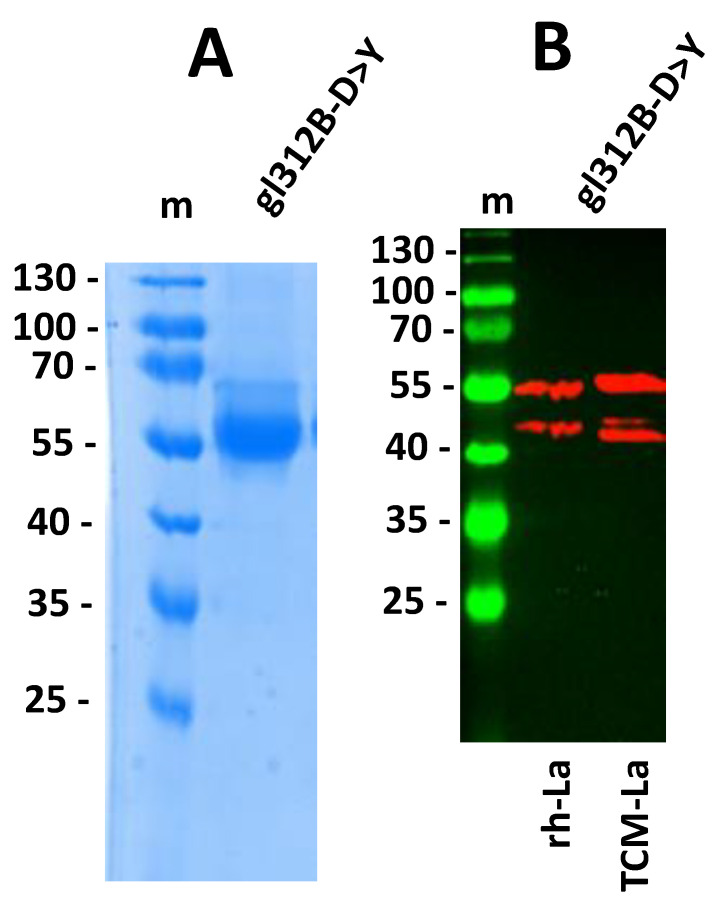
A single aa replacement in the germline sequence restores anti-La reactivity. The SHM causing the mutation of the D residue to a Y residue in the CDR3 region of the V_H_ domain of the 312B ab came into our focus. A 312B IgG4 construct was cloned, in which the D in the germline region was replaced with the Y residue present in the 312B aa sequence. This mutant was termed gl312B-D > Y (germline mutant of 312B in which D is replaced by Y). (**A**) The resulting construct was expressed by permanent 3T3 production cell line. The purified ab gl312B-D > Y was analyzed by SDS-PAGE and stained with Coomassie-Brilliant Blue. (**B**) SDS-PAGE/immunoblot of recombinant human La (rh-La) and the triple cysteine mutant (TCM-La) against the gl312B-D > Y ab. m, protein ladder (kDa).

**Figure 4 ijms-22-12046-f004:**
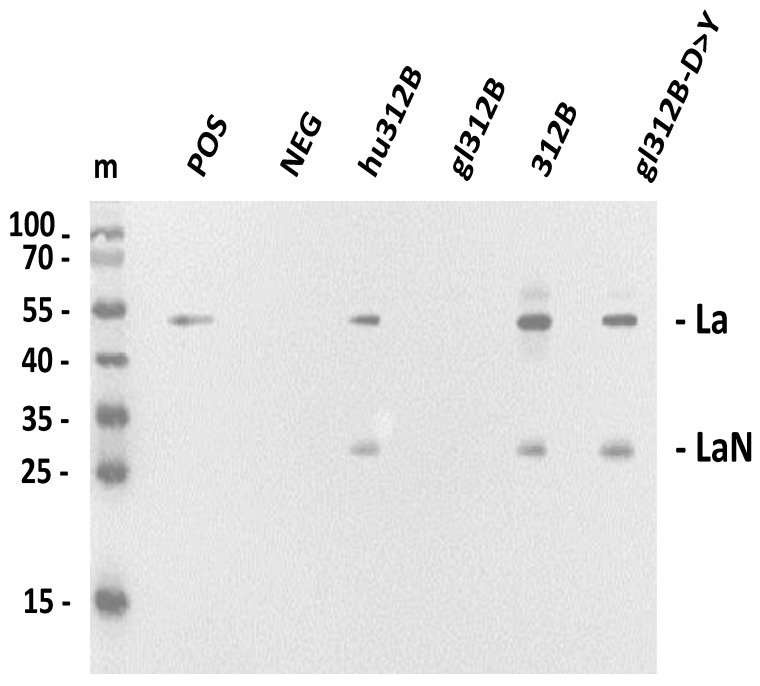
Co-immunoprecipitation of native human La protein. Native human La protein present in total HeLa cell extract (as confirmed in the positive control (POS)) was co-precipitated with the respective 312B ab derivate, including the humanized 312B ab (hu312B); the germline 312B ab, in which all SHMs were mutated back to the germline sequence (gl312B); the mature, murine 312B ab (312B); and the germline ab variant of 312B, in which the aspartate residue in the V_H_ CDR3 region was replaced by a tyrosine residue (gl312B-D > Y). Co-precipitated native human La protein was detected by SDS-PAGE/immunoblotting using the anti-La mab 5B9 and anti-mouse IgG abs-conjugated with peroxidase. POS, positive control (HeLa extract); NEG, negative control (PBS); m, protein ladder; La, human La protein; LaN, N-terminal proteolysis product of La protein.

**Figure 5 ijms-22-12046-f005:**
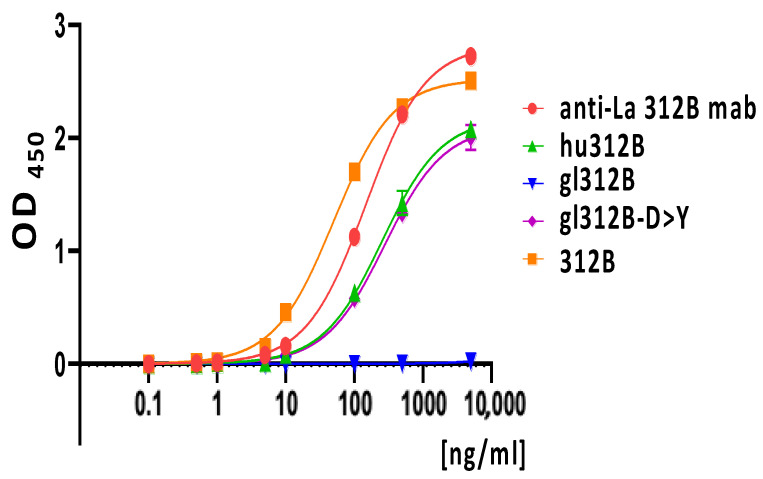
Comparison of the binding curves obtained by ELISA for the different 312B ab derivatives using the TCM-La protein as substrate. Increasing amounts of the respective ab were analyzed ranging from 0.1 to 5.000 ng/mL, including the anti-La mab as secreted from the hybridoma (anti-La 312B mab), the IgG4 construct of the humanized 312B ab (hu312B), the murine 312B ab (312B), the germline ab (gl312B), and the mutant in which the aspartate present in the V_H_ germline sequence is replaced by a tyrosine residue (gl312B-D > Y). Each data point is the result of three estimations.

## Data Availability

The data presented in this study are available on request from the corresponding author.

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
