# Peer review of "A Small Step, a Giant Leap: Somatic Hypermutation of a Single Amino Acid Leads to Anti-La Autoreactivity"

_ijms, 2021, doi:10.3390/ijms222112046_

Round 1

Reviewer 1 Report

Review of ijms-1413538: “A Small Step, A Giant Leap: Somatic Hypermutation of a Single Amino Acid Leads to Anti-La Autoreactivity”

The manuscript written by Bartsch et al. describes the observations that single amino acid replacement in the sequence in the variable heavy chain of antibody leas to the autoreactivity against La protein. 

  • A difficult language together with a difficult topic makes it complicated and unclear, e.g.

“As it recognizes mouse La protein if recombinantly expressed in E. coli the epitope region should be posttranslationally modified in the native mouse La protein [26].”

  • Because “recently” was used twice in the abstract, the reader has the impression that he or she is reading a reference to some earlier research and not currently published.
  • Drawings should be prepared more carefully, e.g. Fig 1B, the molecular marker is for the sample hu312B or not? Why the results are present as separate stipes?
  • Split epitope – I would rather use the nomenclature: discontinouns
  • What is the similarity between mammalian and murine La protein? Maybe graphical representation of this proteins would be helpful to understand the essential regions of the protein.

Author Response

We like to thank both reviewers for their kind comments. As reviewer 2 had no requests for modifications of the ms we just respond to the questions of Reviewer 1. Our answers are given in red. In the ms the modifications are highlighted using the tracking/correction mode of Word.

Reviewer 1:

The manuscript written by Bartsch et al. describes the observations that single amino acid replacement in the sequence in the variable heavy chain of antibody leas to the autoreactivity against La protein. 

  • A difficult language together with a difficult topic makes it complicated and unclear, e.g.

“As it recognizes mouse La protein if recombinantly expressed in E. coli the epitope region should be posttranslationally modified in the native mouse La protein [26].”

We apologize and though we fully agree with the reviewer that we describe a quite complex story in our ms, unfortunately we cannot reduce the complexity of the presented work. In order to facilitate the reading of the ms we have rephrased the above mentioned sentence selected by the reviewer (see page 2, second§, lanes 5 to 11). In addition we tried to simplify further phrases and complex sentences within the ms as indicated by the use of the correction mode of Word.

  • Because “recently” was used twice in the abstract, the reader has the impression that he or she is reading a reference to some earlier research and not currently published.

As requested we have deleted the “recently” and rephrased the abstract to improve the clarity. The modifications of the text are highlighted by the use of the correction mode Word.

  • Drawings should be prepared more carefully, e.g. Fig 1B, the molecular marker is for the sample hu312B or not? Why the results are present as separate stipes?

We looked at the drawings and tried to improve them. Modifications were made as requested to Figure 1B and in addition we al so slightly improved the labelling of one of the axes in figure 5. In detail: Fig. 1B was modified by adding additional dash lines now indicating that the marker lane also belongs to the hu312B lane. We replaced the original Figure 1B with the modified Figure 1B as indicated by the use of the correction mode of Word (see page 4).   

Besides Figure 1B we also slighty corrected the numbering of the concentration axis in Figure 5 by replacing the number “10000” with “10,000”. The modification is indicated by the use of the correction mode of Word (see page 8, Figure 5).   

Indeed, we present the gel as stripes: On the original gel different increasing amounts of the respective proteins were separated in parallel lanes. The additional information is not necessary but simply complicates the writing. For an easy comparison of the purity of the three proteins we decided to select the respective lane showing the highest concentration of each protein sample. So the additional lanes in between were removed resulting in the stripes. For the ms the most important information is the site by site comparison of the germline versus mature 312B construct. So we rearranged these two lanes directly close to each other (shown at the left panel). In contrast, the hu312B protein is less important for the direct comparison. For this reason we positioned it separately besides the marker lanes. We think that separation of the respective information into the three stripes helps to focus on the respective relevant information we want to transfer with these images. Moreover, presenting the three protein samples in the most logical sequence but separately does facilitate their understanding.

  • Split epitope – I would rather use the nomenclature: discontinouns

As requested we have replaced “split epitope” with “discontinuous epitope” (see page 2, §2, lane 4).

  • What is the similarity between mammalian and murine La protein? Maybe graphical representation of this proteins would be helpful to understand the essential regions of the protein.

In order to show the homologies between human and mouse La protein we have prepared a sequence alignment of human and mouse La protein as requested. In this alignment we have schematically included the epitope region recognized by the anti-La mab 312B and the known La domains (La motif, RRM1, and RRM2). This figure was included in the ms as supplemental Figure 1S. We prefer and decided to add it as supplemental Figure as this is an important relevant information which we introduced already in the first sentence of the introduction section (see page 1, Introduction, first sentence). Also these data do not represent essentially new results and are also not directly part of the actual story, namely, how a non reactive B cell turns into an autoreactive one.

Reviewer 2 Report

Bartsch et al further characterise the anti-La/SS-B autoantibody 312B on on which they had previously reported.  Authors isolated recombinant 312B contructs and tested the constructs against human recombinant LA protein (rh-La) and a mutant version which makes La protein insensitive to oxidation.  The authors are humble enough to state the identification of a construct with a replacemnt of aspartate by thyrosine in the CD3R region as "lucky punch" (312B-D>Y). The authors demonstrate by co-immunoprecipitation of native human La protein with different 312B constructs and comparison of binding curves with the respective contructs, in particular with the construct  312B-D>Y that "a single aa replacementcan convert a non-autoreactive B cell to an autoreactive one. 

Author Response

We are very grateful for the reviewer's evaluation. As we were not asked for any modification we have only modified the ms according to the minor requests of reviewer 1.